# Two Teachers are Better Than One: Semi-supervised Elliptical Object Detection by Dual-Teacher Collaborative Guidance

## ABSTRACT

Elliptical Object Detection (EOD) is crucial yet challenging due to complex scenes and varying object characteristics. Existing methods often struggle with parameter configurations and lack adaptability in label-scarce scenarios. To address this, a new semi-supervised teacher-student framework, Dual-Teacher Collaborative Guidance (DTCG), is proposed, comprising a five-parameter teacher detector, a six-parameter teacher detector, and a student detector. This allows the two teachers, specializing in different regression approaches, to co-instruct the student within a unified model, preventing errors and enhancing performance. Additionally, a feature correlation module (FCM) highlights differences between teacher features and employs deformable convolution to select advantageous features for final parameter regression. A collaborative training strategy (CoT) updates the teachers asynchronously, breaking through training and performance bottlenecks. Extensive experiments conducted on two widely recognized datasets affirm the superior performance of our DTCG over other leading competitors across various semi-supervised scenarios. Notably, our method achieves a 5.61% higher performance than the second best method when utilizing only 10% annotated data.

## CCS CONCEPTS

• **Computing methodologies** → **Computer vision**.

## KEYWORDS

Semi-supervised Learning, Elliptical Object Detection, Pseudo-labeling

**ACM Reference Format:**

Anonymous Authors. 2024. Two Teachers are Better Than One: Semi-supervised Elliptical Object Detection by Dual-Teacher Collaborative Guidance. In *Melbourne '24: ACM Symposium on Neural Gaze Detection, 28 October – 1 November, 2024, Melbourne, Australia.* ACM, New York, NY, USA, 9 pages. https://doi.org/XXXXXXX.XXXXXXX

## 1 INTRODUCTION

Human can easily identify elliptical objects from a complex scene. Ellipses can provide us more geometric characteristics, such as rotated angles and shape boundary, which are hardly extracted from simple bounding boxes. In practice, elliptical object detection (EOD) plays a pivotal role in versatile applications spanning various fields such as camera calibration [7], unmanned aerial

**Unpublished working draft. Not for distribution.**

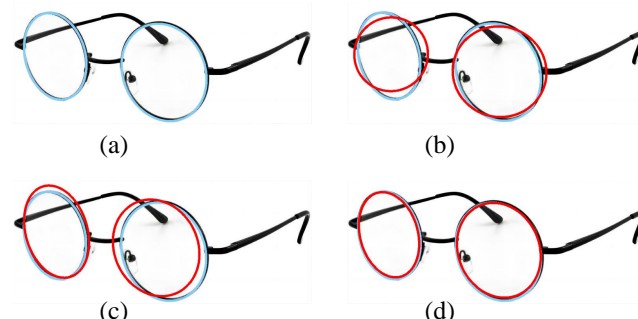

**Figure 1: Comparison of semi-supervised elliptical object detection. The ground-truths and predictions are in blue and red. (a) A raw image. (b) Detection results based on five-parameter-based regression. (c) Detection results based on six-parameter-based regression. (d) Our detection results with combining the two regression approaches. Note that our detected ellipses align with the ground-truths tightly, overcoming the boundary problems of angular regression.**

vehicle (UAV) deployment, and robotic manipulation [4]. However, EOD is still a challenging task due to arbitrary variations such as scales, viewpoints and occlusion. Typically, a line of early works [12, 14, 15, 23, 25] start by detecting a large number of arc segments and then explore their potential combinations to estimate the parameters for the target ellipse. However, these methods suffer from a parameter configuration issue, limiting the detectors to adapt to different variations in real-world scenarios. Following the rapid emergence of generic object detection [8, 20, 30], an increasing number of deep learning based methods [5, 16, 32, 38] are designed for elliptical object detection, resulting in superior performance over traditional methods.

Yet, the top-performing object detectors rely heavily on fully annotated data, but tend to underperform in the context of label-scarcity scenarios. Unlike horizontal bounding boxes, elliptical boxes are more difficult and costly to annotate. Thereby, it is impractical to train an ellipse detector following a fully supervised fashion. Motivated by this problem, semi-supervised object detection (SSOD) [9, 21, 26, 33, 35] has attracted a keen interest because of its capability of dealing with labeled and unlabeled data jointly. A common approach is learning a teacher detector to generate pseudo bounding boxes, with the expectation that the student detector can make consistent predictions on augmented input samples. However, existing methods are dedicated to generic scenes enclosed within horizontal bounding boxes, but are hardly applied to detect various ellipses in images. There has not been enough study about elliptical object detection in a semi-supervised learning paradigm.

Furthermore, ellipses are generally represented with five-parameter regression predicting the angle of an object directly, whereas angular periodicity affects the regression accuracy severely. For example, $-\frac{\pi}{2}$ and $\frac{\pi}{2}$ can represent ellipses of the same angle, making the model confused between the two cases, as shown in Fig. 1(b). For this reason, the work in [43] develops a six-parameter elliptical representation allowing ellipse detection in any direction by representing the angle of the object upon the ellipse focus. Unfortunately, this indirect computational approach leads to inaccurate estimations of the ellipse center, as shown in Fig. 1(c). Hence, we conjecture that there might be a trade-off between five-parameter and six-parameter regression. It is promising yet challenging about how to integrate the advantages from the two regression approaches.

To tackle the problems, this work is the first to address semi-supervised elliptical object Detection (SEOD). To this end, we devise a new semi-supervised teacher-student framework, termed as Dual-Teacher Collaborative Guidance (DTCG). Different from prior work, our DTCG is composed of a five-parameter teacher detector, a six-parameter teacher detector, and a student detector. In this way, the two teachers, specialising in five-parameter and six-parameter regression respectively, can co-instruct the student in a unified model. when a single teacher makes an error, the other teacher might remedy it and prevent the student from being misleading. Moreover, we incorporate the two teachers with a feature correlation module (FCM). Specifically, we highlight the differences between two teacher features through two separate channels, and then utilize deformable convolution to select features advantageous for the final parameter regression. On the other hand, we develop a collaborative training strategy (CoT), which updates five-parameter teacher and six-parameter teacher alternatively by exponential moving average (EMA) approach, breaking through the bottleneck of training and performance. Fig. 1(c) validates the superiority of our method.

To summarize, the contributions of this work are three-fold:

- We pioneer semi-supervised elliptical object detection (SEOD), mitigating annotation costs and enriching the elliptical object detector with unlabeled samples.
- We devise a Dual-Teacher Collaborative Guidance (DTCG) framework, merging the benefits of five-parameter and six-parameter regression through a feature correlation module and asynchronous collaborative training strategy.
- Quantitative and qualitative experiments on two datasets validate the superior performance of our DTCG over competing methods across various semi-supervised scenarios.

## 2 RELATED WORKS

### 2.1 Elliptical Object Detection

**Traditional methods.** Traditional ellipse detection algorithms [13–15, 23, 25] typically use edge detectors to identify arc segments, which are then used to explore potential combinations to estimate the parameters of the ellipses. However, searching for potentially correct arc combinations consumes a lot of execution time. Jia *et al.* propose an efficient arc combination pruning strategy based on a newly developed projection invariant [12]. Likewise, Lu *et al.* propose an edge connection strategy called supporting arcs, achieving high-quality detection results [23]. Nonetheless, identifying continuous arcs from edges is still a challenging task. Parameter

configuration problems often make it difficult for conventional methods adapting to varying sizes of ellipses in real world.

**Deep learning based methods.** Along with the rapid progress of generic object detection [8, 20, 30], current deep learning-based ellipse detection can be broadly categorized into anchor-based and anchor-free methods. Anchor-based ellipse detection methods utilize pre-defined anchor boxes of various sizes and aspect ratios at different positions in the image to accurately localize elliptical objects, such as Ellipse R-CNN [5]. However, due to the necessity of adjusting and classifying numerous anchor points, anchor-based methods suffer from slow inference speed. Instead, anchor-free methods [16, 32, 38] identify objects based on the position of the center, corners of bounding boxes, or key points of the object. They do not rely on prior knowledge of the size, aspect ratio, or position of objects in the image, thereby being more flexible and suitable for ellipse detection. Despite the promising improvement obtained by deep detectors, it cannot conceal their severe degeneration in case of label-scarcity scenarios. *Different from prior approaches, our work proposes to address elliptical object detection effectively via a semi-supervised fashion, thereby effectively resolving the issue of discontinuous angle boundaries in elliptical object detection.*

### 2.2 Semi-Supervised Object Detection

Semi-supervised learning aims to learn representations from labeled and unlabeled samples jointly [2, 31]. In the realm of Semi-Supervised Object Detection (SSOD), a common approach is learning a teacher model to generate pseudo bounding boxes, and expecting the student detector will make consistent predictions on augmented input samples [1, 11, 19, 27, 34, 40, 41]. For instance, STAC [20] designs a hard pseudo-labeling approach generating pseudo-labels on unlabeled data using trained detectors offline. However, the initial predictions from pseudo-labels may involve some noise and limit the detection performance. Subsequently, extensive methods [22, 28, 29, 36, 37] are developed to improve the quality of pseudo-labels. For example, Unbiased Teacher [22] addresses the pseudo-labeling bias using exponential moving average (EMA) and focus loss; SIOD [18] builds a similarity-based pseudo-label generation module; Soft Teacher [44] adaptively weights the loss of each pseudo-label and proposes box dithering to select reliable pseudo-labels; Humble Teacher [28] generates soft labeling objects from the predicted distribution of class probabilities; Dense Teacher [42] introduces a region selection technique to highlight key information while suppressing the noise carried by dense labels. Recently, the work by [10] firstly proposes Semi-supervised Oriented Object Detection (SOOD), and tackle it through dynamically weighing the loss of each pseudo-labeled bounding box based on the angular difference. Nevertheless, there has no work being tailored specifically for Semi-supervised Elliptical Object Detection (SEOD), as elliptical regression is different from that of horizontal bounding boxes. *In this work, we concentrate on SEOD and tackle it with a new dual-teacher collaborative guidance framework, which takes advantage of two elliptical regression approaches effectively.*

## 3 METHODOLOGY

**Overview.** To achieve multi-directional elliptical object detection in real-world scenarios, we propose a semi-supervised elliptical

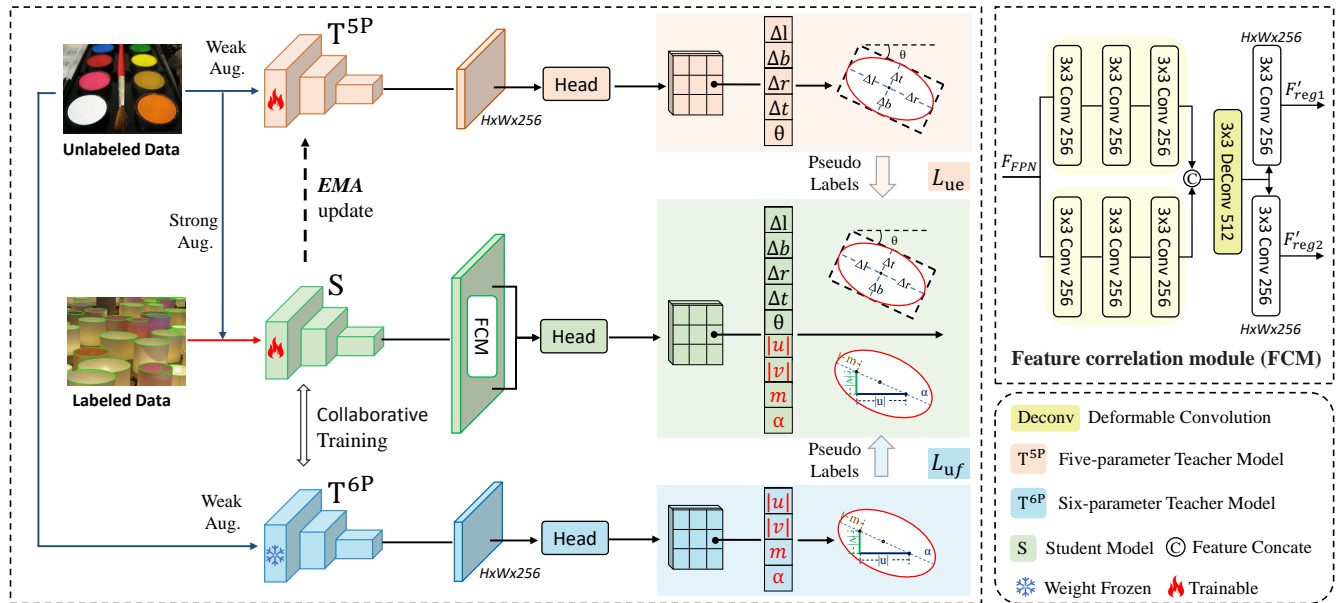

**Figure 2: Overview framework of our dual-teacher collaborative guidance (DTCG) for semi-supervised elliptical object detection. The training data comprises both labeled and unlabeled images. The two teachers, $T^{5P}$ and $T^{6P}$, offer five- and six-parameter pseudo-labels $P_{T^{5P}}$ and $P_{T^{6P}}$, respectively. To enhance the correlation between the knowledge learned from the two teachers, we devise a simple yet effective feature correlation module (FCM). Additionally, we employ an asynchronous updating strategy where the five-parameter teacher updates with the students at each epoch, while the six-parameter teacher updates only at the end of each training period.**

object detection (SEOD) method, which is built on top of the widely used dense pseudo-labeling framework [10]. Our approach, Dual Teacher Collaborative Guidance (DTCG), employs two teacher models proficient in distinct parametric regression methods to jointly guide the student model, as illustrated in Fig. 2. We initiate the training process with fully supervised training on the student model. Subsequently, both teachers are updated by the student model and generate consistent pseudo-labels, albeit represented by different parameters. These pseudo-labels, along with labeled data, are then input into the student model to enhance performance. Notably, the five-parameter model undergoes more frequent updates than the six-parameter model, given its greater stability. Additionally, the asynchronous updating method fosters mutual correction between the two teacher models, preventing simultaneous errors. Furthermore, we design the FCM to establish an intrinsic link between the two teacher branches, as depicted in the top right of Fig. 2.

In this section, we commence by introducing the five-parameter and six-parameter teacher regressions in Sections 3.1 and 3.2, respectively. Following that, in Section 3.3, we elaborate how the dual-teacher guides students. Lastly, in Section 3.4, we delve into the details of the Collaborative Training Strategy (CoT).

## 3.1 Teacher with Five-parameter Regression

The five-parameter regression defines $(x_c, y_c, a, b, \theta)$ to represent an ellipse, where $(x_c, y_c)$ denotes the centre coordinates, $a$ and $b$ are semi-major and semi-minor axes, and $\theta \in [-\frac{\pi}{2}, \frac{\pi}{2})$ is the ellipse orientation [5]. We aim to employ the five-parameter regression

to train the first teacher ($T^{5P}$) on top of a single-stage anchor-free detector [30].

Here, we employ ResNet-50 [9] as the backbone network and simultaneously use a feature pyramid network (FPN) [20] structure. Three feature layers extracted from the backbone network are fed into the FPN to obtain the feature $F_i \in \mathbb{R}^{H \times W \times 256}$ for the regression task. Specifically, for each spatial location $(x, y)$ in the feature map $F_i$, the head predicts both classification scores and regression parameters. The classification scores include the class probability $p_{x,y}^{5P}$ and the normalized distance $s_{x,y}^{5P}$ between the prediction and desired center point. The regression parameters consist of $\Delta l$, $\Delta t$, $\Delta r$, $\Delta b$, $\theta$, as illustrated in the left of Fig. 3. Here, $\Delta l$, $\Delta t$, $\Delta r$, and $\Delta b$ represent the distances from location $(x, y)$ to the object box boundary along the left, top, right, and bottom directions, respectively, while $\theta$ denotes the angle. By decoding these parameters, we can obtain the five parameters of the ellipse by

$$\begin{cases} x_c = x + \cos(\theta) \cdot \dfrac{\Delta r - \Delta l}{2} - \sin(\theta) \cdot \dfrac{\Delta b - \Delta t}{2}, \\ y_c = y + \sin(\theta) \cdot \dfrac{\Delta r - \Delta l}{2} + \cos(\theta) \cdot \dfrac{\Delta b - \Delta t}{2}, \\ a = \dfrac{\Delta r + \Delta l}{2}, \qquad b = \dfrac{\Delta t + \Delta b}{2}, \qquad \theta = \theta. \end{cases} \quad (1)$$

Note that, the task of ellipse detection is treated as a binary classification problem, where the ellipse is regarded as the foreground, and the remaining is background otherwise. For the predicted value $p_{x,y}^{5P}$, we use the category corresponding to the value with high probability as the classification result. A position $(x, y)$ is considered

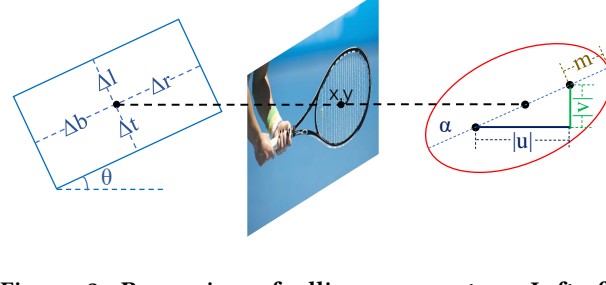

**Figure 3: Regression of ellipse parameters. Left: five-parameter regression using $(\Delta l, \Delta t, \Delta r, \Delta b, \theta)$. Right: six-parameter regression including $(x, y, |u|, |v|, m, \alpha)$.**

a positive sample if it falls within any ground truth box and shares the same class label as the ground truth instance. Conversely, if the position does not meet these criteria, it is treated as a negative sample, representing the background class. In addition to the label for classification, we require the following parameters: $\delta^* = (x_c^*, y_c^*, a^*, b^*, \theta^*)$ and $s^*$, which is defined as:

$$s_{x,y}^* = \sqrt{\frac{\min(\Delta l^*, \Delta r^*)}{\max(\Delta l^*, \Delta r^*)} \times \frac{\min(\Delta t^*, \Delta b^*)}{\max(\Delta t^*, \Delta b^*)}}, \quad (2)$$

where $(\Delta l^*, \Delta t^*, \Delta r^*, \Delta b^*)$ denote the real distance to the object boundary, which is obtained by inverting $\delta^*$ as Eq. 1. The $s^*$ is used for centrality determination, which is intended to suppress low-quality detection boxes that deviate from the target center. Eventually, the loss of training the five-parameter teacher model combines three terms including classification loss, regression loss, and centre-point loss:

$$\mathcal{L}_{T^{5P}} = \frac{1}{N_{pos}} \sum_{x,y} \mathcal{L}_{cls}(p_{x,y}^{5P}, p_{x,y}^*) + \frac{1}{N_{pos}} \sum_{x,y} \mathcal{L}_{reg}(\delta_{x,y}^{5P}, t_{x,y}^*)$$
$$+ \frac{1}{N_{pos}} \sum_{x,y} \mathcal{L}_{ctr}(s_{x,y}^{5P}, s_{x,y}^*),$$
$$(3)$$

where $\mathcal{L}_{cls}$ is Focal loss , $\mathcal{L}_{reg}$ is RotatedIoU loss and $\mathcal{L}_{ctr}$ is Binary Cross Entropy loss. $N_{pos}$ denotes the number of positive samples.

## 3.2 Teacher with Six-parameter Regression

Due to the potential discontinuity in angle boundaries resulting from the aforementioned five-parameter regression, we adopt a six-parameter approach to represent an ellipse [43], namely $(x_c, y_c, |u|, |v|, m, \alpha)$. This method avoids direct prediction of object angles by encoding them into vectors. Fig. 3 illustrates how the six-parameter strategy decomposes the object angle into two components, $(|u|, |v|)$, of the focal point vector. Specifically, when the object resides in quadrants one and three, both $(|u|, |v|)$ are either positive or negative; whereas in quadrants two and four, one of $(|u|, |v|)$ is positive and the other negative. Additionally, when the object aligns with the axes, one component equals zero. The parameters $(x_c, y_c)$ denote the ellipse center, $m$ represents the difference between the semi-major axis length and the semi-focal distance. Moreover, $\alpha$

indicates the orientation relative to the axes, with $\alpha = 1$ for quadrants one and three or axis alignment, and $\alpha = 0$ for quadrants two and four.

We employ a single-stage anchor-free detection network for six-parameter regression, identical to the approach used in the five-parameter model [30]. Similar to the five-parameter teachers, at each location $(x, y)$, we regress the classification probability and the corresponding parameters. In this scenario, the classification probability remains consistent with that of the five-parameter teacher. The regression parameters comprise $(|u|, |v|, m, \alpha)$, from which we derive the six-parameter elliptic box $\delta^{6P} = (x_c, y_c, |u|, |v|, m, \alpha)$ as depicted in the right of Fig. 3. We normalize $\delta^{6P}$ for loss calculation. The normalisation process is as follows:

$$\begin{cases} x_c = \dfrac{x}{s}, & y_c = \dfrac{y}{s}, & |u| = \log(|u|/s + \text{eps}), \\ |v| = \log(|v|/s + \text{eps}), & m = \log(m/s + \text{eps}), \end{cases} \quad (4)$$

where $eps$ is a very small number and $s$ is the maximum value of the width and height of the input image. Here again, our overall loss is Eq. 3. Note that $\mathcal{L}_{cls}$ and $\mathcal{L}_{ctr}$ are consistent with the definition of Eq. 3, but $\mathcal{L}_{reg}$ is Smooth $L_1$ loss.

## 3.3 Student with Dual-teacher Guidance

Direct angle prediction based on the five-parameter regression tends to encounter issues with angle periodicity and boundary discontinuities, while the six-parameter approach may struggle with convergence and inaccurate center estimation. To address these challenges, we employ two teacher models to jointly guide a student model in generating both five and six parameters, facilitated by a feature correlation module and joint regression.

Typically, the five-parameter teacher $T^{5P}$ and the six-parameter teacher $T^{6P}$ receive weakly augmented unlabeled images, while the student receives strongly augmented unlabeled images and labeled ones. In this way, the student undergoes both supervised and unsupervised training. For the supervised part, the student is trained using the labelled data and a supervised loss $\mathcal{L}_{sup}$. For the unsupervised part, we leverage the consistent positive predictions of the teachers $T^{5P}$ and $T^{6P}$ to generate the pseudo-labels $P_{T^{5P}}$ and $P_{T^{6P}}$. Notably, we select the student's prediction $P_s$ at the same location and the two teachers for consistency constraints to form unsupervised losses $\mathcal{L}_{u^{5P}}$ and $\mathcal{L}_{u^{6P}}$. Finally, the overall loss function is defined as:

$$\mathcal{L}_u = \sum_i^N \mu_i^{rot}(\mathcal{L}_{cls}(p_i^S, p_i^T) + \mathcal{L}_{reg}(\delta_i^S, \delta_i^T) + \mathcal{L}_{ctr}(s_i^S, s_i^T)),$$
$$\mathcal{L}_{total} = \mathcal{L}_{sup} + \omega_u(\mathcal{L}_{u^{5P}} + \lambda \mathcal{L}_{u^{6P}}),$$
$$(5)$$

where $\mathcal{L}_{reg}$ uses L1 Loss, $\mathcal{L}_{cls}$ and $\mathcal{L}_{ctr}$ are aligned with Eq. 3, $\mu_i^{rot}$ uses parameters consistent with [10], $\omega_u$ represents the unsupervised loss weights, and $\lambda$ is the hyperparameter that balances the effects of the two teachers. We employ an asynchronous update strategy for two teachers, which is detailed in Section 3.4. Next, we delve into the specifics of constructing the student.

**Feature Correlation Module.** To better integrate the knowledge learned from the two teachers, we propose a feature correlation module (FCM) based on the two regression branches. Within the FCM, we decouple and recombine the features in student model,

**Figure 4: Collaborative training strategy.** $T^{5P}$ **and** $T^{6P}$ **denote five- and six-parameter teachers, respectively, and** $S$ **denotes the student model. At each stage,** $T^{5P}$ **is obtained from the EMA of** $S$**; at the end of each stage, the weights of** $T^{6P}$ **and** $S$ **are exchanged.**

and the resulting features are used in the regression of the two tasks separately. This module establishes a correlation between the two branches and facilitates mutual guidance and information exchange between the branches.

Concretely, the five-parameter and six-parameter regression features, $F_{reg1} \in \mathbb{R}^{H \times W \times 256}$ and $F_{reg2} \in \mathbb{R}^{H \times W \times 256}$, respectively, are extracted using two independent sets of convolutional layers:

$$\begin{cases} F_{reg1} = f_{conv}^{reg1}(F_{FPN}), \\ F_{reg2} = f_{conv}^{reg2}(F_{FPN}), \end{cases} \quad (6)$$

where $f_{conv}^{reg1}(\cdot)$ and $f_{conv}^{reg2}(\cdot)$ denote two independent sets of three 3x3 convolutional layers. The relevant features $F_{reg1}$ and $F_{reg2}$ for the five-parameter and six-parameter regression tasks are generated jointly. To enhance the interaction between the two features, we concatenate the features $F_{reg1}$ and $F_{reg2}$ together to form $F_{reg1-reg2}$ containing the two regression features. To align the five-parameter and six-parameter regression tasks, we apply a $3 \times 3$ deformable convolution [3] to the concatenated $F_{reg1-reg2}$. We then use a deformable convolution with a kernel size of $3 \times 3$ to learn the offsets of the regression features. The aligned features $F_{reg1}$ and $F_{reg2}$ are expressed as follows:

$$\begin{cases} F'_{reg1} = f_{dcn}^{reg1}(F_{reg1-reg2}), \\ F'_{reg2} = f_{dcn}^{reg2}(F_{reg1-reg2}), \end{cases} \quad (7)$$

where $f_{dcn}^{reg1}(\cdot)$ and $f_{dcn}^{reg2}(\cdot)$ represent standard deformable convolution operations, allowing the utilization of correlated features $F'_{reg1}$ and $F'_{reg2}$ for regression tasks involving both five- and six-parameter boxes.

**Joint Regression** Our objective is to harness the benefits of both teachers, thereby enhancing the detection precision in object detection. As shown in Fig. 3, we present a detailed process of joint regression. Specifically, at each spatial location $(x, y)$ in the feature map, we conduct both five-parameter and six-parameter regressions to generate nine parameters. The first five parameters ($\Delta l$, $\Delta t$, $\Delta r$, $\Delta b$, $\theta$) are defined in the five-parameter-based

---

**Algorithm 1** Training of Dual-teacher Collaborative Guidance.

---

1: DYNAMIC_INTERVAL = 1
2: STATIC_INTERVAL = 3200
3: # Inner-period training update strategy
4: **if** current_iter mod DYNAMIC_INTERVAL == 0 **then**
5:     # Augmentation of unlabeled image data to form batch data
6:     img_w ← weak_aug(unlabel_images)
7:     img_s ← strong_aug(img_w)
8:     # Get the pseudo-label prediction pair
9:     pred_s ← student(img_s)
10:    pseudo_st ← six_teacher(img_w)
11:    pseudo_ft ← five_teacher(img_w)
12:    # Calculation of unsupervised losses
13:    loss_st ← consistency_regularization(pred_s, pseudo_st)
14:    loss_ft ← consistency_regularization(pred_s, pseudo_ft)
15:    loss ← $\lambda*$loss_st + loss_ft
16:    # Update the student by back-propagation
17:    loss.backward( )
18:    # Update the five teacher by EMA
19:    update_teacher(student, five_teacher)
20: **end if**
21: # Outer-period exchange of teacher-student
22: **if** current_iter mod STATIC_INTERVAL == 0 **then**
23:    exchange_weight(student, six_teacher)
24: **end if**

---

regression. We employ Eq. 1 at each position $(x, y)$ to generate final five parameters $\delta' = (x_1, y_1, a_1, b_1, \theta_1)$. The remaining four parameters ($|u|, |v|, m, \alpha$) are delineated in the six-parameter-based regression and can also be decoded into another set of five parameters $\delta'' = (x_2, y_2, a_2, b_2, \theta_2)$. The decoding process for the six parameters is outlined as follows:

$$\begin{cases} x_2 = x, \qquad a_2 = \dfrac{\sqrt{u^2 + v^2}}{2} + m, \\[2mm] y_2 = y, \qquad b_2 = \sqrt{a_2^2 - \dfrac{u^2 + v^2}{4}}, \\[2mm] \theta_2 = \begin{cases} \arcsin\left(\dfrac{|u|}{\sqrt{u^2+v^2}}\right), & \text{if } \alpha = 1 \lor |u| = 0 \lor |v| = 0, \\[2mm] -\arcsin\left(\dfrac{|u|}{\sqrt{u^2+v^2}}\right), & \text{if } \alpha = 0. \end{cases} \end{cases} \quad (8)$$

The five parameters may lead to inaccurate angle predictions due to boundary discontinuities and angle periodicity. Conversely, the six parameters present challenges in convergence, making it arduous to regress to precise bounding boxes. Nevertheless, the six parameters offer more accurate angle predictions compared to the five parameters. Consequently, we combine the bounding box derived from the five parameters with the angle derived from the six parameters. The resulting composite is denoted as $\delta = (x_1, y_1, a_1, b_1, \theta_2)$.

## 3.4 Collaborative Training Strategy

Most existing training strategies for semi-supervised object detection rely on a self-training paradigm using the mean teacher (MT) framework [29]. In this way, the teacher model is an exponential

Table 1: Quantitative comparison on GED and SmartPhone datasets, under the settings of 10%, 20%, and 30% labelled data. In addition, we show the average scores across the three settings. P, R, F-M denote the precision, recall and F-measure metrics.

| Dataset | Method | 10% | | | 20% | | | 30% | | | Average | | |
|---|---|---|---|---|---|---|---|---|---|---|---|---|---|
| | | P | R | F-M | P | R | F-M | P | R | F-M | P | R | F-M |
| GED | Dense Teacher [42] | 65.41 | 59.17 | 62.21 | 73.71 | 63.94 | 68.48 | 71.35 | 63.76 | 67.34 | 70.16 | 62.29 | 66.01 |
| | SOOD [10] | 68.39 | 60.34 | 64.11 | 75.27 | 65.21 | 69.88 | 76.42 | 68.08 | 72.01 | 73.36 | 64.54 | 68.67 |
| | **DTCG (Ours)** | **72.10** | **64.69** | **68.19** | **76.53** | **66.38** | **71.10** | **77.95** | **69.35** | **73.41** | **75.52** | **66.74** | **70.9** |
| SmartPhone | Dense Teacher [42] | 71.35 | 63.76 | 67.34 | 74.30 | 64.97 | 69.32 | 82.12 | 71.01 | 76.16 | 75.92 | 66.58 | 70.94 |
| | SOOD [10] | 77.38 | 62.80 | 69.33 | 77.39 | 64.49 | 70.36 | 81.84 | 68.59 | 74.63 | 78.87 | 65.29 | 71.44 |
| | **DTCG (Ours)** | **78.27** | **67.87** | **72.70** | **80.75** | **67.87** | **73.75** | **89.94** | **75.60** | **82.15** | **82.99** | **70.45** | **76.2** |

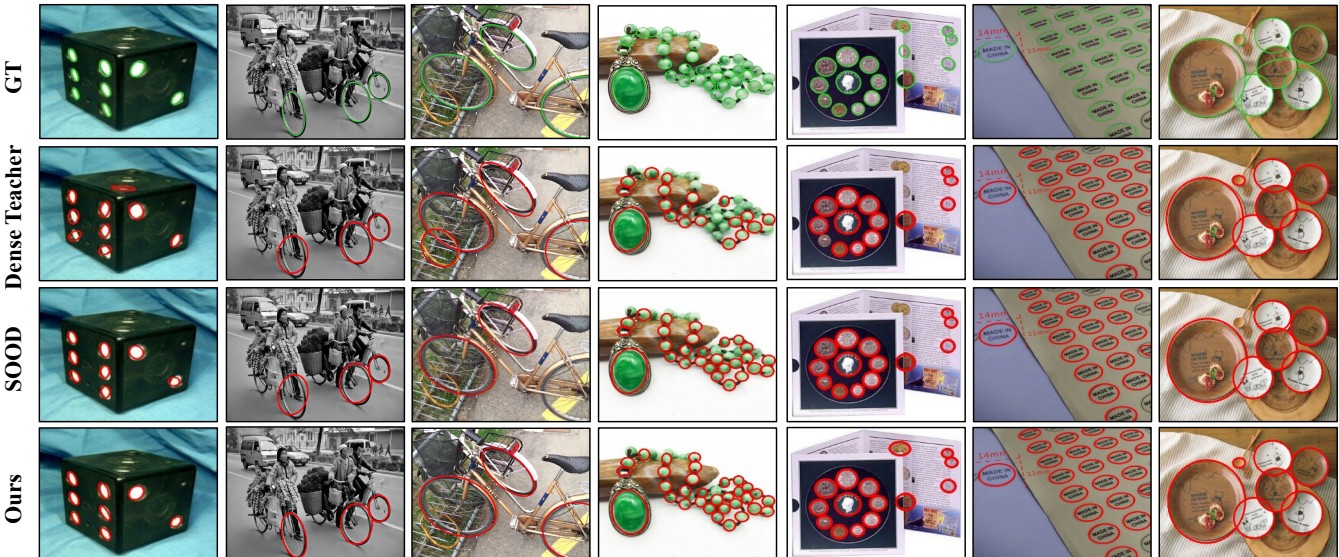

GT  Dense Teacher  SOOD  Ours

Figure 5: Qualitative comparison with image samples from the GED dataset.

moving average (EMA) of the student model at different time steps, and the student model is updated based on pseudo-labels provided by the teacher. However, this paradigm makes the teacher model vulnerable to the cumulative error caused by the student model, leading to potential instability and a performance bottleneck.

To address this problem, we develop a collaborative training strategy (CoT) as shown in Fig. 4, which can be divided into two phases: (1) **Inner-period dynamic updates:** during each period, the weights of the six-parameter teacher model remain static and constant, while the five-parameter teacher model is dynamic and updated using the EMA of the student model. (2) **Outer-period static update:** at the end of each training period, the weights are exchanged between the student model and the six-parameter teacher model. In other words, the roles of the six-parameter teacher and student are swapped at the end of each training period.

This updating strategy based on CoT offers the following benefits. **Student Model:** the statically updated teacher model serves as a performance lower bound, ensuring stability for the student model. In case of issues with dynamic teacher guidance, swapping reverts the student model to its previous state, effectively preventing rapid performance deterioration and enhancing robustness.

**Teacher model:** the static update strategy ensures regular knowledge updates for the six-parameter teacher model, promoting stability. Additionally, the five-parameter static teacher integrates past student model exchanges, reducing noise and enhancing resilience compared to traditional average teacher frameworks. This approach prevents catastrophic forgetting and uncontrollable crashes, leading to improved detection performance. Additionally, we describe the whole training procedure of our DTCG model in Algorithm 1.

## 4 EXPERIMENTS

### 4.1 Datasets and Experiment Settings

**Dataset protocols.** We conduct extensive experiments on two popular ellipse detection benchmarks: General Ellipse Detection (GED) [32] and Smartphone [6]. The former comprises 629 images captured from six video cameras, featuring traffic signs and bicycles from various perspectives. The latter is composed of 1443 images, which are collected from real-world scenes. Since there is no research on semi-supervised elliptical object detection, we thereby follow the protocols used for semi-supervised object detection [10, 42]. In practice, we randomly select 10%, 20% and 30% of

**Table 2: Ablation study on two teachers, feature correlation module (FCM) and collaborative training (CoT), with the setting of 10% labeled data in the GED dataset.**

| Model | $T^{5P}$ | $T^{6P}$ | FCM | CoT | P | R | F-M |
|-------|------|------|-----|-----|-------|-------|-------|
| M1 | √ | | | | 68.39 | 60.34 | 64.11 |
| M2 | | √ | | | 47.32 | 40.29 | 43.52 |
| M3 | √ | √ | | | 70.37 | 62.04 | 66.10 |
| M4 | √ | √ | √ | | 71.88 | 64.28 | 67.86 |
| M5 | √ | √ | √ | √ | **72.10** | **64.69** | **68.19** |

the images in the training set as labeled data, and the remaining images act as unlabeled data.

**Evaluation metrics.** We adhered to common quantitative metrics for evaluating ellipse detection, including Precision, Recall, and F-Measure. The threshold applied to determine True Positives (the count of correctly detected ellipses) is commonly set with 0.8.

**Implementation details.** Without loss of generality, we adopt FCOS [30] as a representative anchor-free detector and utilize ResNet-50 [9] and FPN [20] as the backbone for building our DTCG model. Besides, asymmetric data augmentation is applied to the unlabeled data, including weak augmentation with random flipping [17, 39], and strong augmentation via random flipping, color jitter, random grayscale, and random Gaussian blur [4, 28]. Inspired by previous SSOD work [22, 42], we employ a "burn-in" strategy for initializing the two teacher networks in our DTCG. The model is trained for 120,000 iterations on a single NVIDIA RTX 4090 GPU. The SGD optimizer is used with an initial learning rate of 0.0025, reduced by a factor of 10 at 80k and 110k iterations. Momentum and weight decay are set to 0.9996 and 0.0001, respectively. The pseudo-label sampling rate is set to 0.25 by default.

## 4.2 Main Results

As there is no study on semi-supervised elliptical object detection (SEOD), to the best of our knowledge, we instead choose two top-performing semi-supervised object detectors: Dense Teacher [42] and SOOD [10]. Particularly, SOOD is the first to propose addressing semi-supervised oriented object detection, which is related to elliptical object detection. Given the publicly released source codes, we re-implement these two baseline methods with GED and SmartPhone datasets. Below, we elaborate on the quantitative and qualitative results.

**Quantitative comparison.** From the results reported in Table 1, we can see that the proposed DTCG attains state-of-the-art performance across various settings (*e.g.* 10%, 20%, 30% labeled data) in both datasets. To be more specific, DTCG outperforms SOOD consistently on GED dataset, especially with an average increase of 2.16 Precision, 2.2 Recall, 2.23 F-Measure scores, respectively. Considering the fact that SmartPhone is less challenging than GED, the improvements achieved by DTCG thereby become more remarkable, including 4.12% Precision, 5.16% Recall, 4.76% F-Measure scores on average. The results verify the effectiveness of our DTCG model tailored for SEOD.

**Qualitative comparison.** In addition to the quantitative results above, we further carry on a qualitative comparison, as shown in Fig. 5. Overall, DTCG offers more accurate boundaries and angles than Dense Teacher [42] and SOOD [10], exhibiting a reduction

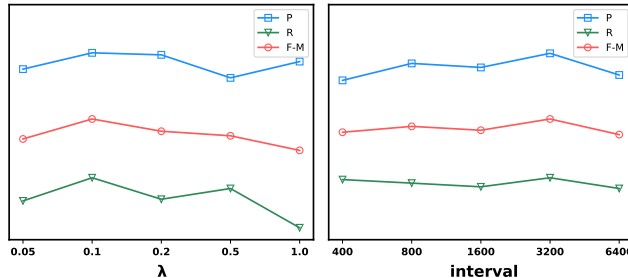

**Figure 6: Impact of loss weight $\lambda$ and outer-period interval given the GED dataset with the setting of 10% labeled data.**

of prediction error and an enhancement of detection quality. Concretely, in the first and second columns of Fig. 5, both Dense Teacher and SOOD predict several wrong angles, which can seriously affect the detection accuracy; for the third column, Dense Teacher produces a false positive (*i.e.* "bicycle lock"), while SOOD misses a target ellipse of "bicycle wheel". Besides, it can be seen that our method remains effective even for some dense scenes including the fourth, fifth and sixth columns. In terms of failure cases, all the methods still fail to detect heavily occluded targets such as the "big plate" in the seventh column.

## 4.3 Ablation Study

We conduct ablation experiments to elaborate on the effectiveness of our DTCG model. All the experiments are performed on the GED dataset using 10% labeled data.

**Component analysis.** This results in Table 2 shed more light on the effectiveness of key components in DTCG. First of all, by comparing the results of M3 with M1 and M2, we confirm the fact that combining two teachers (*i.e.* $T^{5P}$ and $T^{6P}$) outperforms any single teacher. As six-parameter teacher is good at accurate angle regression, the dual-teacher model is able to be free from the wrongly predicted angles caused by five-parameter regression. We note that the M2 model based on $T^{6P}$ only performs poorly because of regressing inaccurate ellipse centers. Then, feature correlation module (FCM) helps to boost the performance with a consistent increase of 1.51% Precision, 2.24% Recall, and 1.76% F-measure scores (refer to M3 and M4). Furthermore, collaborative training strategy (CoT) is conducive to mutual learning between the two teachers, making M5 (*i.e.* the full DTCG model) perform better.

**Impact of loss weight $\lambda$.** This experiment aims to study the impact of the loss weight $\lambda$ on the two teachers ($T^{5P}$ and $T^{6P}$). The results with varying $\lambda$ are presented in Fig. 6(a). We can see that the optimal performance is obtained when $\lambda$ is set to 0.1, where Precision is 72.10%, Recall is 64.69%, and F-Measure is 68.19%. When we change $\lambda$ to 0.05, all the scores decrease remarkably, due to a loss of correct angles predicted by $T^{6P}$. On the other hand, increasing $\lambda$ may involve more low-quality and noisy pseudo-labels caused by $T^{6P}$. In one word, we find that $T^{5P}$ plays a more important role in guiding the student than $T^{6P}$. However, this should not overlook the beneficial knowledge learned from $T^{6P}$.

**Impact of outer-period interval.** In this experiment, we aim to analyze the impact of the outer-period interval (epoch) on DTCG.

**Table 3: Cross-dataset evaluation with GED and SmartPhone.**

| Labeled | Unlabeled | Method | P | R | F-M |
|---------|-----------|--------|---|---|-----|
| GED | SmartPhone | Dense Teacher[42] | 48.67 | 53.14 | 50.81 |
| | | SOOD [10] | 66.76 | 59.66 | 63.01 |
| | | **DTCG (Ours)** | **71.01** | **64.49** | **67.59** |
| SmartPhone | GED | Dense Teacher[42] | 49.05 | 38.36 | 43.05 |
| | | SOOD [10] | 47.50 | 39.12 | 43.10 |
| | | **DTCG (Ours)** | **60.81** | **39.66** | **48.01** |

Figure 6(b) presents the settings for different intervals. We observe that the best performance is achieved when the interval is set to 3200, reaching to 72.10% Precision, 64.69% Recall, and 68.19% F-Measure. We speculate that this interval effectively fine-tunes the model for optimal convergence. Adjusting the interval either upwards or downwards has a noticeable decrease on the performance.

## 4.4 Cross-dataset Evaluation

In general, existing works assume that both labeled and unlabeled samples are from the same dataset. However, this is not always feasible in real-world applications. To further validate the generalization ability of the model, we conduct a cross-dataset experiment using both GED and SmartPhone, whose results are summarised in Table 3. One the one hand, we merge the labeled training set of GED with the unlabeled training set of SmartPhone. Then we train the model with the merged training set and evaluate its performance on the SmartPhone test set. As a result, DTCG achieves 71.01% Precision, 64.49% Recall, and 67.59% F-Measure, which excels SOOD with a large margin of 4% on average. On the other hand, we train with the labeled training set of SmartPhone and the unlabeled training set of GED, and evaluate the performance on the GED test set. Likewise, the compared results prove the superiority of DTCG over other competitors. This study suggests that DTCG offers promising generalization ability even when labeled and unlabeled samples follow different data distributions.

## 4.5 Fully-supervised Setting

DTCG is tailored specifically for SEOD, whereas it potentially acts as a general framework for fully supervised setting as well. To this end, we evaluate DTCG given all the samples are fully labeled. with no loss of generality, we follow the same implementation details, except removing the collaborative training strategy.

**Quantitative comparison.** As reported in Table 4, we compare with state-of-the-art fully-supervised methods, including traditional methods CNED [12], ArcLs [7], CM [13] and Meng [24], and deep learning methods ElDet [32] and FCOS [30]. For GED, our method achieves an improvement of 6.58% Precision, 4.72% Recall, and 4.48% F-measure over prior state-of-the-art. In terms of SmartPhone, we achieve the best performance on Recall and F-measure scores, except that our Precision is lower than that of ElDet [32]. Ultimately, these results demonstrate that our method is equally effective for fully supervised setting, thanks to solving the angular periodicity and boundary issues.

**Qualitative comparison.** In addition to the quantitative results above, we further carry on a qualitative comparison, as shown in Fig. 7. Overall, our method provides more accurate boundaries and angles than ElDet [32] and FCOS [30]. Concretely, in the first

**Table 4: Fully supervised results on the GED and SmartPhone datasets. The best and second-best results are in bold and underlined, respectively.**

| Method | GED | | | SmartPhone | | |
|--------|-----|-----|-----|-----|-----|-----|
| | P | R | F-M | P | R | F-M |
| CNED[12] | 35.86 | 47.39 | 40.83 | 62.82 | 52.66 | 57.29 |
| ArcLs[7] | 54.19 | 52.51 | 53.33 | 79.28 | 69.33 | 73.96 |
| CM[13] | 53.91 | 49.74 | 51.74 | 81.36 | 66.43 | 73.14 |
| Meng[24] | 35.57 | 26.57 | 30.42 | 83.85 | 67.45 | 74.76 |
| ElDet [32] | 72.73 | 62.78 | 67.39 | **90.55** | 60.15 | 72.28 |
| FCOS [30] | 77.82 | 68.13 | 73.72 | 82.47 | 69.32 | 75.33 |
| **DTCG (Ours)** | **84.40** | **72.85** | **78.20** | 87.00 | **69.56** | **77.32** |

column of Fig. 7, both ElDet and FCOS miss the target ellipse; in the second and third columns, our detection quality is better than that of Eldet and FCOS; for the fourth column, ElDet predicts the wrong angle while FCOS misses the ellipse target of (*i.e.* "table").

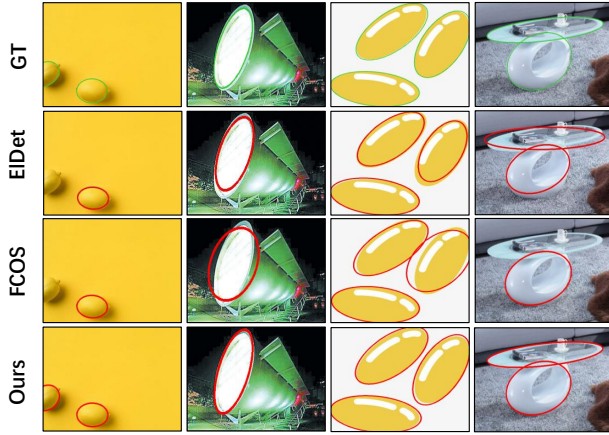

**Figure 7: Qualitative comparison with image samples from the GED dataset in a fully supervised setting.**

## 5 CONCLUSION

In this paper, we introduce a novel semi-supervised solution for elliptical object detection. We propose a Dual-Teacher Collaborative Guidance (DTCG) framework that integrates the strengths of five-parameter and six-parameter regression through the Feature Correlation Module (FCM) and asynchronous Collaborative Training Strategy (CoT). The framework aims to address the angular boundary discontinuity problem caused by angular periodicity. Quantitative and qualitative experiments on two datasets validate the superior performance of our DTCG over competing methods across various semi-supervised scenarios. This highlights the efficacy of our method in addressing the challenges of semi-supervised elliptical object detection. Despite achieving satisfactory results in semi-supervised elliptical object detection, our method still has shortcomings. Limited by the post-processing operation (NMS), it can be easily filtered out when the elliptical object is heavily occluded.

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

Received 20 February 2007; revised 12 March 2009; accepted 5 June 2009

