# OpenReview forum: "Two Teachers Are Better Than One: Semi-supervised Elliptical Object Detection by  Dual-Teacher Collaborative Guidance"
_acmmm.org/ACMMM/2024/Conference — MM2024 Poster_

### Official Review · Reviewer_7he4 · 2024-05-20

**Rating:** 4
**Confidence:** 4

**Summary:**

The manuscript proposes a new semi-supervised elliptical object detection framework DTCG, which takes full advantage of five-parameter and six-parameter models. Specifically, the framework contains two teachers and utilizes a feature correlation module (FCM) and collaborative training strategy (CoT) to generate high-quality pseudo labels for student models. The experiments reveal the effectiveness of the proposed framework and components.

**Strengths:**

- Novelty and technical correctness: The authors discuss the advantages and disadvantages of five-parameter and six-parameter models, and find a trade-off between two types of models naturally.
- Adequate evaluation: The authors do extensive experiments on fully-supervised and semi-supervised learning.
- Clarity: Most descriptions are clear.
- Applications: The method is significant to apply to elliptical object detection in the real world.

**Limitations:**

1. The symbols such as 𝐹_{𝐹𝑃𝑁} in Eq.6, and functions such as consistency_regularization() in Algorithm 1 are not defined in the text.
2. In Table 2, what is the weight-updating strategy of T^6p  without CoT? What is the effectiveness without the outer-period static update, i.e., only updating the weight of T^6p at the end of each training period?

**Suitability:**

3

---

### Official Review · Reviewer_G5KH · 2024-05-23

**Rating:** 2
**Confidence:** 3

**Summary:**

This paper introduce a semi-supervised paradigm for elliptical object detection, which includes a five-parameter teacher detector, a six-parameter teacher detector, and a student detector.  The two teacher models generate pseudo-labels for the student model. The student model has two regression tasks to conduct both five-parameter and six-parameter regressions to generate nine parameters. A feature correlation module (FCM) is proposed to decouple and recombine features for the two downstream regression tasks separately. A collaborative training strategy is proposed to update the parameters of teacher models via the inner-period dynamic update and the outer-period static update.

**Strengths:**

1. This paper introduces a semi-supervised paradigm for elliptical object detection and states that this is the first attempt.
2. This paper is well organized and the experimental results are significant.

**Limitations:**

1. The writting of this paper needs to be further improved. For example, so many “ing” clause in this paper, which degrades the fluency of paper. Besides, there are some minor grammatical errors, e.g., the word “when” should be “When” at line 138, and the inconsistent singular and plural description bewteen “Inner-period dynamic updates” (line 625) and  “Outer-period static update”(line 627).
2. Some descriptions are not clear, such as “breaking through the bottleneck of training and performance” at line 147. How to understand the bottleneck? Please provide examples.
3. Some confusions in Fig.2.  The five-parameter teacher model is updated by EMA and the six-parameter teacher model is updated by parameter copying. However, Fig. 2 depicts that five-parameter teacher model is trainable while the six-parameter teacher model is frozen.
4. Some questions about why the five-parameter teacher model uses EMA while the six-parameter teacher model uses the  parameter coping. How about the manners of paramenter update of these two teacher models are exchanged.
5. How to implement Dense Teacher and SOOD to elliptical object detection, as they are designed for general object detection and orientated object detection, respectively. Details need to be introduced.
6. The research of this paper seems to be a computer vision related work and not so stay close to the ACMMM theme.

**Suitability:**

1

---

### Official Review · Reviewer_tLxC · 2024-05-25

**Rating:** 5
**Confidence:** 2

**Summary:**

Elliptical Object Detection (EOD) is challenging due to complex scenes and varying object characteristics, with existing methods often struggling in label-scarce scenarios. To address this, a new semi-supervised framework, Dual-Teacher Collaborative Guidance (DTCG), is proposed, featuring two teacher detectors and a student detector that collaboratively enhance performance. A feature correlation module (FCM) improves feature selection for parameter regression, and a collaborative training strategy (CoT) updates the teachers asynchronously. Extensive experiments demonstrate DTCG's superior performance, achieving a 5.61% higher performance than the second-best method with only 10% annotated data.

**Strengths:**

1. This paper is well motivated.

2. This paper is easy to follow.

3. The performance of the proposed method is good.

**Limitations:**

1. Figure 2 and Figure 4 seem conflict to each other. In Figure 2, T^{6P} is fixed but in Figure 4, T^{6P} is upated.

**Suitability:**

2

---

### Official Review · Reviewer_gqUv · 2024-05-27

**Rating:** 5
**Confidence:** 4

**Summary:**

This paper focuses on Semi-Supervised Elliptical Object Detection to deal with label-scarce scenarios. Considering that existing methods struggle with parameter configurations, this paper proposes a new teacher-student semi-supervised learning framework, Dual-Teacher Collaborative Guidance (DTCG), which comprises a five-parameter teacher and a six-parameter teacher to guide student training complementarily. Specifically, this paper designs a feature correlation module (FCM) to assist dual-teacher learning, and a collaborative training strategy to update dual-teacher asynchronously. Experiments validate the effectiveness of the proposed method.

**Strengths:**

-	This is the first work to study Semi-Supervised Elliptical Object Detection.
-	This paper designs an interesting dual-teacher framework with complementary parameter configurations for Semi-Supervised Elliptical Object Detection.
-	The proposed method is well motivated.

**Limitations:**

-	Paper structure. In the section of method, both Sec.3.1 and Sec.3.2 should be moved out as a section of background knowledge about five-parameter and six-parameter elliptical object detection. The section of method should focus more on the design of semi-supervised learning.
-	Why update five-parameter teacher via EMA and update six-parameter teacher via exchanging the weights from student and teacher? What about exchanging the ways to updates five-parameter teacher and six-parameter teacher? I cannot find out from the ablation study.
-	The proposed dual-teacher framework and the update manner to swap teacher and student during training are quite similar to [1]. Both aim to mitigate the instable training problem. Discussion and comparison with [1] are necessary from the perspective of teacher-student learning technique.

[1] Periodically Exchange Teacher-Student for Source-Free Object Detection. ICCV 2023.

**Suitability:**

2

---

### Meta-Review · Area_Chair_JCfw · 2024-07-01

**Recommendation:** Accept (Poster)
**Confidence:** 5

**Metareview:**

This paper proposes an elliptical object detection approach based on the semi-supervised teacher-student framework. Notable experimental improvements are achieved using only 10% labeled data. The paper initially received 2 weak accepts, 1 borderline accept and 1 weak reject. Reviewers raised concerns on the presentation issues, the methodological details, relation with existing works, etc. After rebuttal, most of these concerns are addressed, while reviewer G5KH still thinks the paper is not suitable to the scope of ACM MM. The AC believes computer vision ideas are important to the multi-media area, and the method proposed in this paper should be interesting to the ACM MM attendees. Therefore, the AC suggests to accept this paper.